# Liquid Biopsy as a Means of Assessing Prognosis and Identifying Novel Risk Factors in Multiple Myeloma

**DOI:** 10.3390/ijms26178505

**Published:** 2025-09-01

**Authors:** Maiia Soloveva, Maksim Solovev, Igor Yakutik, Bella Biderman, Elena Nikulina, Natalya Risinskaya, Tatiana Obukhova, Maria Gladysheva, Alla Kovrigina, Yulia Chabaeva, Sergei Kulikov, Andrey Sudarikov, Larisa Mendeleeva

**Affiliations:** National Medical Research Center for Hematology, Novy Zykovski Lane, 4a, 125167 Moscow, Russia; maxsolovej@mail.ru (M.S.); igorya90@list.ru (I.Y.); bella_biderman@mail.ru (B.B.); lenysh2007@rambler.ru (E.N.); risinska@gmail.com (N.R.); obukhova@blood.ru (T.O.); makislitsyna@gmail.com (M.G.); kovrigina-alla@mail.ru (A.K.); uchabaeva@gmail.com (Y.C.); kulikov.s@blood.ru (S.K.); dusha@blood.ru (A.S.); mendeleeva.l@blood.ru (L.M.)

**Keywords:** multiple myeloma, liquid biopsy, ctDNA, plasmacytoma, *NRAS*, *KRAS*, *BRAF*, array-CGH

## Abstract

Multiple myeloma (MM) is a complex genetic disease characterized by the heterogeneity of tumor cells. We have measured *KRAS*, *NRAS*, and *BRAF* gene mutations in circulating free tumor DNA (ctDNA) from plasma, bone marrow, and plasmacytoma samples as well as their correlation with various clinical and laboratory parameters. The prospective study included 113 MM patients (74 with plasmacytoma and 39 without), treated at the National Medical Research Center for Hematology (Moscow, Russia) from 2009 to 2024. FISH was performed on CD138+ bone marrow cells for 104 patients and array-CGH for two extramedullary plasmacytoma samples. Mutation analysis on CD138+ bone marrow cells was performed for 99 patients, on ctDNA for 80 patients, and, in 26 cases, samples of plasmacytoma were also investigated. Mutations in the *KRAS*, *NRAS*, and *BRAF* genes either in bone marrow, ctDNA, or plasmacytoma samples were found in 50% of patients. In patients with plasmacytoma, mutations in ctDNA were found in 28% of cases versus 0% in cases without plasmacytoma (*p* = 0.0007). Rare “noncanonical” *KRAS* and *NRAS* gene mutations were also more frequent in ctDNA compared to the bone marrow substrate (50% versus 9%, *p* = 0.01). Liquid biopsy in MM, particularly identification of the *KRAS*, *NRAS*, and *BRAF* gene mutations in ctDNA, is a valuable instrument for prognostication. Researching the intricate mechanisms underlying extramedullary involvement, and identifying novel high-risk factors associated with the disease, is worthwhile.

## 1. Introduction

Multiple myeloma (MM) is a malignant lymphoproliferative disease, the substrate of which is a tumor plasma cell. MM is a genetically complex disease characterized by the spatial heterogeneity of tumor clones [1,2]. In some patients, plasma cells acquire the ability to proliferate independently outside the bone marrow, forming plasmacytomas. The mechanisms driving this autonomous growth remain unclear. Plasmacytomas are classified into two types: bone plasmacytoma is more common, which occurs as a result of bone destruction and the tumor going beyond its limits; a rare type of lesion is extramedullary plasmacytoma, which forms due to the hematogenous dissemination of a tumor cell into any organs and tissues, without association with bone [3,4,5]. The results of plasmacytoma studies are more often presented in small samples of MM patients [6,7,8]. The difficulties in studying the pathogenesis of plasmacytoma are due to several reasons; one is that extramedullary manifestations of MM are not a frequent event. In some cases, the anatomical location of the plasmacytoma or the severity of the patient’s condition create technical difficulties for performing a biopsy. In addition, a plasmacytoma biopsy is not included in the MM diagnostic algorithm—histological confirmation of a plasmacytoma is not required if there is no doubt about the diagnosis and bone marrow damage. At the same time, in real clinical practice, the biopsy of the lesion is sometimes the first stage for verifying the plasma cell nature of the tumor. DNA extracted from paraffin-embedded plasmacytoma biopsies is a valuable resource for studying the molecular and genetic features of these lesions. Mutations in the *KRAS*, *NRAS*, and *BRAF* genes have long been studied in MM, with their frequency increasing from monoclonal gammopathy of undetermined significance (MGUS) to advanced disease stages [9,10]. However, published data on the impact of RAS–ERK pathway mutations on disease course are contradictory. There are studies indicating an unfavorable prognostic value of mutations in the *KRAS* gene and a decrease in sensitivity to proteasome inhibitors with mutations in the *NRAS* gene [10,11]. Another study, however, demonstrated the positive effect of the mutated status of these genes on treatment outcomes in MM patients [12]. The ambiguous results in early studies are probably due to small patient samples and treatment regimens involving only a single targeted drug [12,13,14]. More and more publications are devoted to the issues of adding BRAF, MEK, or MTOR inhibitors to standard MM therapy for targeting the tumor cell [15,16,17]. Liquid biopsy is a rapidly advancing field in oncology research, although the method is not new—DNA circulating in plasma began to be studied in the middle of the last century. A lack of standardization in the methodology also limits the use of liquid biopsy. In MM, a niche for the use of tumor ctDNA analysis is actively being explored. The ability to determine the spatial and temporal heterogeneity of a tumor, measure the tumor load without invasive procedures, and avoid the need for special preparation of the patient, before taking blood, is only part of the advantages of a liquid biopsy. In MM, bone marrow aspirate is a traditional tool for studying genetic disorders in tumor cells and assessing risk. However, the multifaceted spatial heterogeneity of MM cannot be fully represented by just one bone marrow sample. Attempts are being made to use ctDNA to study the mutational profile of genes in dynamics, as a predictor of antitumor response and in the assessment of minimal residual disease [18,19,20]. The aim of this work was to identify mutations in the RAS–ERK genes in different MM substrates (ctDNA, bone marrow, plasmacytoma), using various cytogenetic and molecular methods, and to assess the association of mutation data with different disease conditions.

## 2. Results

### 2.1. Analysis of the KRAS, NRAS, and BRAF Gene Mutations

The frequency of the *KRAS*, *NRAS*, and *BRAF* gene mutations in any of the tumor substrate (bone marrow, ctDNA, plasmacytoma) in the entire group was 50% (56 out of 113). We analyzed the frequency of high-risk cytogenetic aberrations in bone marrow, depending on the presence of mutations in MAP kinase genes, at one of the three tumor loci. The analysis included 104 patients with FISH data available. The differences are unreliable; however, there is a tendency towards a rarer detection of high-risk aberrations in patients with the *KRAS*, *NRAS*, and *BRAF* gene mutations at any tumor locus. Thus, in the group of patients with mutations (*n* = 53), the proportion of patients with high cytogenetic risk was 45% (*n* = 24). In the group without mutations (*n* = 51), a high risk was determined in 63% of cases (*n* = 32), *p* = 0.08. The data are shown in Figure 1.

The main part of this study was the assessment of mutations in RAS–ERK cascade genes in different tumor loci from the same patients, and comparison of the results depending on the presence of plasmacytomas. Paired tumor samples (CD138+ bone marrow cells and ctDNA in plasma) were studied in 80 patients, 50 with plasmacytomas and 30 without. In addition, the plasmacytoma substrate was analyzed in 11 patients. Figure 2 shows a diagram of this part of the study.

Table 1 shows the characteristics of 80 MM patients with and without plasmacytomas.

As can be seen from Table 1, there were no significant differences between the groups in terms of most clinical and laboratory parameters; however, a significant difference was noted in the frequency of stage III D-S assignment. Table 2 shows data on *KRAS*, *NRAS*, and *BRAF* gene mutations, or their combination, detected in bone marrow and the ctDNA of patients with and without plasmacytomas. It should be noted that *NRAS* and *BRAF* mutations in the bone marrow of patients with plasmacytomas are detected somewhat more often than in patients without plasmacytomas, but the differences are not significant. A key finding was that mutations in these genes within ctDNA were exclusively detected in the plasmacytoma group. The frequency of the detection of mutations in the RAS–ERK cascade genes in the ctDNA was 17.5% (14 out of 80).

Appendix A clearly demonstrates the mutation profile of paired tumor samples from patients with and without plasmacytomas.

We assigned the patients to a group “with mutations in RAS–ERK cascade genes” if mutations were detected in one of the three genes. Figure 3 represents mutations in the RAS–ERK cascade genes in patients with plasmacytomas (*n* = 50) and without plasmacytomas (*n* = 30) in two tumor locations. It turned out that patients with plasmacytomas were significantly more likely to have mutations in ctDNA compared with patients without plasmacytomas (28% vs. 0, *p* = 0.0007). Specifically, mutations were identified in the ctDNA of 14 out of 50 patients in the plasmacytoma group, but in none of the 30 patients without plasmacytoma. In contrast, the frequency of mutations in the bone marrow did not differ significantly between the two groups.

Most of the mutations in the *KRAS* and *NRAS* genes affected the “classic” codons 12, 13, and 61. We also noted rare, “non-classical” variants of mutations in the *KRAS* and *NRAS* genes (e.g., A59G, V29A, L19F, and K88E of *KRAS*; L95P, Y64N, A83V, and S87I (G260T) of *NRAS*). Of the 80 bone marrow samples, mutations in the *KRAS* and *NRAS* genes were detected in 32 cases (29 patients, 3 patients simultaneously had two mutations: in different codons of one gene, or single mutations in two genes). At the same time, “classical” codons were noted in 29 cases (91%), and “nonclassical” codons in 3 cases (9%). Of the 80 ctDNA samples, mutations in the *KRAS* and *NRAS* genes were detected in 10 observations, with an equal ratio of “classical” and “nonclassical” codons. Thus, rare mutations in the *KRAS* and *NRAS* genes were significantly more common in tumor ctDNA compared with the bone marrow substrate (50% vs. 9%, *p* = 0.01), Figure 4. Notably, these unusual mutations were exclusively detected in patients with plasmacytoma.

In the 11 cases where archival biopsy material was available, the mutation status of genes was also investigated in the plasmacytoma substrate. Mutations were found in six patients, and the results are shown in Figure 5.

Figure 5 illustrates the biological and anatomical heterogeneity of MM. The data reveal distinct mutational patterns across different tumor sites within the same patients. In four cases (Patients 1, 2, 4, and 6), a mutation was detected exclusively in the plasmacytoma sample and was absent from the paired bone marrow and/or ctDNA. Conversely, in the other two mutation-positive cases, the mutation found in the plasmacytoma was also present in at least one other site. Furthermore, in some instances (e.g., Patient 3 in bone marrow; Patient 5 in ctDNA), mutations were identified in other substrates but not in the plasmacytoma itself. Although the sample size is limited (*n* = 11), these findings indicate that the frequency of mutations in plasmacytoma samples was 55% (6/11). Notably, in the majority of these mutation-positive plasmacytomas (4 out of 6, 67%), the mutation was unique to the plasmacytoma and not shared with other tumor sites.

### 2.2. Analysis of the Clinical Course of MM Complicated with Plasmacytoma

We expanded the group using archival material from plasmacytoma biopsies from 24 MM patients. DNA was isolated from 26 samples of soft-tissue components. Two simultaneous plasmacytoma were presented in two patients, one bone and one extramedullary. In most patients (20 out of 24), the plasmacytoma was biopsied at the onset of MM, and, in four patients, at the relapse stage. Twenty-six samples included 15 bone plasmacytomas and a unique collection of 11 extramedullary plasmacytomas. Figure 6 shows the localization of plasmacytoma. Vertebrae and skull bones were a frequent localization for bone plasmacytoma. The spectrum of extramedullary lesions was diverse: stomach, retroperitoneal space, mammary gland, tonsil, tongue, and tissues from other localizations.

The overall frequency of mutations in the RAS–ERK cascade genes within the expanded cohort of 26 plasmacytoma samples was 42% (11 out of 26 samples). Analysis of the 15 bone plasmacytomas revealed mutations in the *KRAS* gene in 27% of cases (*n* = 4), in the *NRAS* gene in 6.7% (*n* = 1), and in the *BRAF* gene in 6.7% (*n* = 1). In the 11 extramedullary plasmacytomas, mutations were detected in the *NRAS* gene in 27% of cases (*n* = 3) and in the *BRAF* gene in 18% (*n* = 2), suggesting a tendency for *KRAS* mutations to be more frequent in bone lesions, while *NRAS* mutations were more common in extramedullary sites. It was noted that “classical” codons were more often affected in the plasmacytoma substrate, in contrast to the ctDNA. So, out of 11 cases, only one unusual mutation was found: T50I in the *NRAS* gene.

The diagram in Figure 7 schematically shows the follow-up time for 24 patients whose plasmacytoma biopsies were examined. In a number of patients, the follow-up period is not yet long enough; they are undergoing program treatment. In patients with a long follow-up period, it is possible to track how long they have not received therapy. Death was recorded in 10 patients; nine deaths resulted from MM progression and one was due to complications in an allogeneic hematopoietic stem cell (HSC) transplantation. Notably, the majority of these fatal events occurred early after diagnosis—within the first two years, with two cases within the initial 6 months. At the same time, it should be emphasized that patients received adequate antitumor therapy, and, in all possible cases, high-dose methods were used, such as autologous and allogeneic HSCs transplantation.

In Patient 12, four tumor locations were studied: bone marrow, ctDNA, and the substrate of bone and extramedullary plasmacytoma. There were no mutations in any of the RAS–ERK cascade genes in any of the four tumor locations, despite the fact that, according to our observations and the literature, the detection of mutations is a fairly common event. In Patient 24, the *NRAS* gene was mutated in the plasmacytoma substrate, but no mutation of this gene was detected in the bone marrow and ctDNA. Both patients had a rare extramedullary type of plasmacytoma—in the soft tissues of the chest in Patient 12 and in the retroperitoneal space in Patient 24. We performed molecular karyotyping of these two extramedullary cases using the array-CGH method. Notably, plasmacytoma biopsies were performed at the onset of MM. The chest wall plasmacytoma from Patient 12 exhibited the most pronounced aberrations, with numerous chromosomal abnormalities of varying sizes affecting all chromosomes except 12, 18, and 20 (see Appendix A). In the same patient, a rare event was observed: chromothripsis of the long arm of chromosome 3 (see Figure 8).

Array-CGH analysis of the retroperitoneal extramedullary plasmacytoma from Patient 24 revealed aberrations in nine chromosomes: 1, 4, 8, 10, 11, 16, 17, 22, and X (see Appendix A).

The molecular karyotype lesions of both extramedullary plasmacytoma were often localized on extended sections of chromosomes. Thus, in both plasmacytomas, a copy number neutral loss of heterozygosity (cnLOH) was detected, affecting the entire long arm of chromosomes 10 and 16. In one case, a deletion and cnLOH of the entire short arm of chromosome X were noted and, in another case, a monosomy of chromosome X. The two plasmacytomas shared identical genomic alterations at seven distinct chromosomal loci (Table 3). Lesions of sizes less than 4.5 Mb were not analyzed.

Patient 12, who presented with an extramedullary plasmacytoma of the chest soft tissues and exhibited the most complex molecular karyotype, including chromothripsis of chromosome 3q, was classified as having triple-hit myeloma by FISH. Thus, del17p13, t (14;16), 1q amplification, *CDKN2C*/1p32 gene locus deletion, trisomy 8, and c-*MYC*/8q24 duplication were detected in the patient’s bone marrow. Despite high-dose chemotherapy followed by lenalidomide maintenance, the patient succumbed to disease progression 23 months after diagnosis. Patient 24, with a retroperitoneal extramedullary plasmacytoma, was identified as having double-hit myeloma. FISH detected t (11;14), del17p13, 1q21 amplification, 8q24 rearrangement, and biallelic *CDKN2C*/1p32 deletion in the bone marrow sample. This case followed a fulminant course, resulting in mortality just 1.5 months after initial MM diagnosis.

## 3. Discussion

The association between the mutated status of MAP kinase genes in MM and high-risk cytogenetic aberrations continues to be investigated. In a 2008 publication, it was shown that mutations in RAS–ERK cascade genes are infrequently combined with such unfavorable factors as t(4;14) and 17p13 deletion. At the same time, the relationship between the mutated status of these genes and the presence of t(11;14) was noted. The authors concluded that t(11;14), combined with a mutation of MAP kinase genes, is associated with unfavorable survival rates, although this translocation itself is a marker of the standard risk [10]. An integrative network data analysis published in 2018 shows a link between mutations in the *NRAS* gene and hyperdiploidy, and the absence of an association with 1q amplification. A rare combination of a mutation in the *NRAS* gene with t(4;14) translocation was also noted. The results suggested that a mutation in the *NRAS* gene is a favorable prognostic factor [14]. According to our data from 104 cases, there was a tendency towards rare detection of high-risk aberrations in patients with mutations in the RAS–ERK cascade genes at any tumor locus compared with patients without mutations (45% vs. 63%, *p* = 0.08).

The study of the RAS–ERK signaling cascade in MM is of great practical importance, given the emergence of new targeted drugs. The aspects of the effectiveness of BRAF/MEK inhibition in relapsed/refractory MM, and the search for mechanisms of tumor cells acquiring resistance to treatment with BRAF inhibitors, are relevant.

One limitation of this study is the absence of an analysis of the antitumor response in relation to the mutational status of the RAS–ERK genes. This limitation stems from the heterogeneous nature of the included patient cohort, which comprised both candidates and non-candidates for high-dose therapy, and the non-uniform administration of induction regimens. Future analysis of therapy effectiveness is planned for a larger, more uniformly treated cohort. A principal finding of this work is that mutations in *KRAS*, *NRAS*, or *BRAF* genes within ctDNA were significantly more prevalent in patients with plasmacytoma compared to those without (28% vs. 0, *p* = 0.0007). This leads us to propose that the detection of ctDNA mutations could serve as a valuable tool for predicting disease course and identifying novel risk factors in MM. This premise is supported by growing global research efforts in liquid biopsy. Thus, M. Vlachova et al., using mass spectrometry on peripheral blood samples from MM patients, built a model that predicts primary extramedullary lesions with high sensitivity and specificity [21].

Liquid biopsy is considered to be a non-invasive alternative to traditional tissue biopsy methods. We propose that liquid biopsy can become an additional tool for studying the pathogenesis of plasmacytoma. Having a rare collection of plasmacytoma biopsies from MM patients, we managed to investigate the molecular and genetic features of the tumors. There is a tendency for the *KRAS* gene to be more frequently mutated in the bone plasmacytoma, whereas the *NRAS* gene is more frequently mutated in extramedullary lesions. However, we acknowledge that these observations require cautious interpretation due to the limited sample size of our cohort.

Here we report that non-canonical *KRAS* and *NRAS* mutations were detected significantly more frequently in ctDNA than in bone marrow samples (50% vs. 9%, *p* = 0.01; Figure 4). Notably, all these rare *KRAS* and *NRAS* gene mutations were found only in the group of patients with plasmacytoma. Earlier, we noted that patients with mutations in “non-classical” codons have a variety of factors associated with unfavorable prognosis [22]. Accumulation and further analysis of data are necessary to understand this phenomenon. We believe that liquid biopsy in MM should also be used to search for new high-risk factors, such as *KRAS* and *NRAS* mutations that affect “non-classical” codons.

In two patients, molecular karyotyping has shown common copy number aberrations at loci 1p34, 1q21, 10q11, 16q23, 17p13, Xp22, and Xq11. Over time, genetic lesions accumulate from minimal changes in MGUS to a variety of aberrations during the progression of the disease to secondary plasma cell leukemia. This usually takes years. However, array-CGH data indicate that the accumulation of genetic aberrations can occur very quickly. What event triggers the rapid transformation of plasma cells into malignancy remains unknown.

Chromosome 1 rearrangement was detected in the substrate of two plasmacytoma according to array-CGH. FISH method also confirmed the amplification of 1q and deletion of the *CDKN2C*/1p32 gene locus in the bone marrow of both patients. Amplification of 1q21 is a high-risk aberration that occurs at the onset of MM in more than 30% of patients [23]. The 1q21 region contains many genes involved in MM pathogenesis, such as *CKS1B*, *BCL9*, *ANP32E*, *MCL1*, *PSMD4*, *IL6R*, *ILF2*, and *ADAR* [24,25]. The product of the *PSMD4* gene is a subunit of the 19S proteasome complex, a component of the 26S proteasome. Studies have shown that, with a high expression of the *PSMD4* gene, worse survival rates were observed due to a decrease in the antitumor effect of proteasome inhibitors, and the level of *PSMD4* expression correlated with the number of 1q21 copies [26,27]. In two patients, we noted a deletion of the 1p34 region, which includes the most important genes involved in cell cycle arrest: *FAF1*, and *CDKN2C*. The loss of genes involved in apoptosis is one of the causes of the uncontrolled proliferation of tumor cells [28,29]. The frequent detection of chromosome 1 rearrangements in MM is due to the fact that about 30% of “high-risk genes” are located on this chromosome [30]. The *NRAS* gene, a mutation of which has been detected in a retroperitoneal plasmacytoma, but not in bone marrow or plasma ctDNA, is located on chromosome 1 in the 1p13.2 region. In the same plasmacytoma, we observed a complex rearrangement of chromosome 8—duplication of the long and short arms. FISH analysis of the bone marrow confirmed an aberration at the 8q24 locus. Various abnormalities of chromosome 8 have been observed in more than a third of patients with MM at the onset, and are more frequently detected in patients with advanced stages of the disease, and are characterized by an adverse effect on prognosis [31,32]. It is known that the genes of the superfamily of tumor necrosis factor receptors (*TRAIL-R1, TRAIL-R2*) involved in cell apoptosis are located on the short arm of chromosome 8 [33]. The most important proto-oncogene, *MYC*, is located on the long arm of chromosome 8 in the 8q24.1 region. The gene product is a transcription factor involved in the processes of cell growth, proliferation, and apoptosis [34,35]. The 8q region also contains genes (*RRM2B*, *NOV*, *RAD21*) associated with the metastatic process, which is well described in studies on the progression of colorectal cancer [36,37]. We report the deletion of the entire long arm of chromosome 10 (locus 10q11.21q26.3) in the substrate of two extramedullary plasmacytoma. The tumor suppressor gene *PTEN*, a product of the phosphatase gene, which is an inhibitor of the PI3K/AKT signaling pathway, is located in region 10q23.31. An inactivating mutation or deletion involving the *PTEN* gene locus led to hyperstimulation of this signaling pathway, resulting in exponentially increased cell division [38]. Both cases we report also had a deletion of the entire long arm of chromosome 16. In the 16q23.2 region, the *MAF* proto-oncogene, which encodes the eponymous transcription factor, is located [39,40]. In a patient without mutations in the RAS–ERK cascade genes, we have found a rare phenomenon: chromothripsis of the long arm of chromosome 3 in the substrate of an extramedullary plasmacytoma. Chromothripsis is the destruction of a single chromosome (or part of it), followed by chaotic repair. According to published data, this phenomenon forms the basis of complex chromosomal rearrangements that occur in 2–3% of all cancers [41]. F. Magrangeas et al., analyzing samples of 764 MM patients, revealed signs of chromothripsis in 1.3% of patients (*n* = 10) [42]. Rearrangements of chromosomes 1q, 2, 3, 8q, 10, and 16q have been described, and the adverse effect of this phenomenon on the prognosis of MM has been shown. In one case, the authors noted more than 50 chromosome exchanges in the 16q region. In another later study on a representative sample of patients with MM (*n* = 752), a catastrophic event, chromothripsis, was shown in 24% of patients [43]. The value of studying this phenomenon is growing, given the fact that chromothripsis may be associated with extremely poor outcomes of the disease. In this regard, the important role of assessing the molecular karyotype of a tumor by the array-CGH is revealed.

## 4. Materials and Methods

A single-center prospective study from 2009 to 2024 included 113 patients (48 men and 65 women) with newly diagnosed MM, aged 29–83 years (median 55 years), among whom 74 patients had plasmacytoma and 39 did not. The diagnosis was established in accordance with the IMWG criteria, and performing the full range of necessary laboratory and instrumental investigations. A low-dose CT scan of the entire body was performed to assess the bone system. The clinical and laboratory parameters of patients determined at the onset of the disease are presented in Table 4. More than half of the patients (54%) have high-risk cytogenetic aberrations. In 65% of patients, the disease occurred with the presence of plasmacytoma. Bone plasmacytoma prevailed in 80% of cases, extramedullary plasmacytomas were noted in 7% of patients, and a combination of two types of plasmacytoma was found in another 13% of cases.

Positive immunomagnetic selection of CD138+ bone marrow cells, using a monoclonal antibody to CD138 (STEMCELL Technologies, Vancouver, BC, Canada), was performed in 104 patients according to the manufacturer’s protocol. FISH examination of CD138+ cells using DNA probes was performed to detect translocations of 14q32/IgH, 8q24/MYC, deletions of 17p13/TP53, 13q14, 1p32, and amplifications of 1q21 and multiple trisomies (Wuhan HealthCare Biotechnology, Wuhan, China). When t(4;14), t(14;16), del17p13, and 1q21 amplifications were detected, the patient was classified as a high cytogenetic risk group. Comparative genomic hybridization on a microarray (array-CGH) was performed on DNA from two extramedullary plasmacytomas. DNA isolation from plasmacytoma was performed by phenol–chloroform extraction. Array-CGH was performed using a high-density micromatrix specially developed for hematological diseases by OGT 8x60k + SNP (Oxford Gene Technology, Oxford, UK) using the Agilent platform (Agilent Technology, Santa Clara, CA, USA). The reference female DNA for analysis was supplied by OGT (UK). Copy number variations (CNVs) were determined based on measurements of the signal intensity from probes from the reference, and test DNA was hybridized on the slide. The hg37 version was used as the reference genome. The data were processed using CytoSureTM Interpret Software Version: 4.11 (Cytosure Cancer SNP Arrays, haematological cancer + SNP-8 × 60 k, OGT, Oxford, UK). Pathogenic variations in the number of DNA copies were evaluated in the DECIPHER v11.0 clinical database.

In 99 patients, the *KRAS*, *NRAS*, and *BRAF* gene mutations were measured using genomic DNA isolated from CD138+ bone marrow cell samples; in 80 patients, ctDNA samples were examined in parallel. In addition, 26 plasmacytomas were also analyzed, using DNA isolated from archival material of tumor biopsies. The mutational status of the *KRAS* and *NRAS* genes was studied by high-throughput sequencing (MiSeq, Illumina, San Diego, CA, USA), and the findings were confirmed by Sanger sequencing (Nanophor 05, Institute of Analytical Instrumentation of RAS, Moscow, Russia). The *BRAF* V600E mutation was determined by real-time allele-specific PCR (CFX96 Touch, Bio-Rad, Hercules, CA, USA). The design of the study is shown in Figure 9.

Standard methods of descriptive statistics and frequency analysis were used to analyze the data. To test hypotheses about differences in the distribution of the categorical features between groups, an analysis of contingency tables was performed using the two-way Fisher’s exact test to assess statistical significance.

## 5. Conclusions

In routine multiple myeloma (MM) practice, stratification into risk groups is still based on the examination of a bone marrow sample. However, given the spatial heterogeneity of MM, there is a need to study additional tumor substrates for proper disease stratification and prognostication. The presence of mutations in the *KRAS*, *NRAS*, and *BRAF* genes is more prevalent in the ctDNA of patients with plasmacytoma compared to those without. A higher frequency of “non-classical” mutations in the ctDNA compared to the bone marrow sample is shown. Mutations in RAS–ERK cascade genes are detected in approximately half of cases of newly diagnosed MM, regardless of tumor location. Liquid biopsy can be used to predict the course of a disease, study the pathogenesis of extramedullary lesions, and to search for new high-risk factors. A wide range of molecular genetic research methods (FISH, array-CGH, targeted sequencing) is needed to search for key events triggering the mechanism of malignancy of a tumor plasma cell.

## Figures and Tables

**Figure 1 ijms-26-08505-f001:**
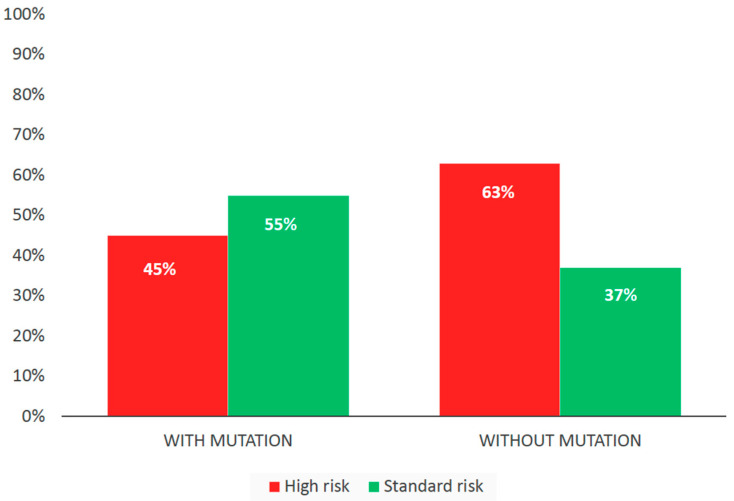
Cytogenetic risk group depending on the mutation status of MAP kinase genes in MM patients.

**Figure 2 ijms-26-08505-f002:**
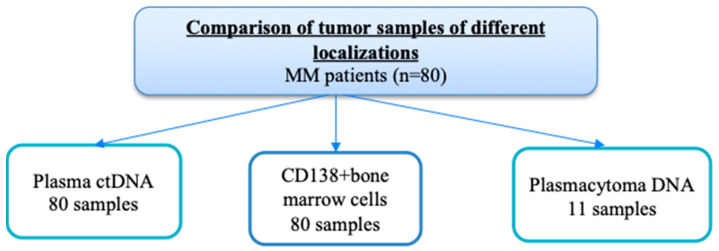
Comparison of paired tumor samples.

**Figure 3 ijms-26-08505-f003:**
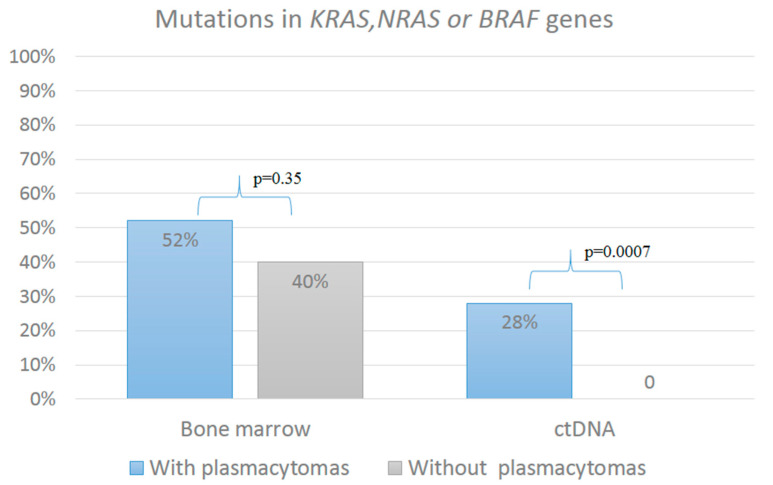
Analysis of mutations in the RAS–ERK pathway genes in bone marrow and ctDNA, depending on the presence of plasmacytoma. Mutations in bone marrow were detected with approximately the same frequency in patients with and without plasmacytomas. Mutations in ctDNA were significantly more frequently detected in patients with plasmacytomas.

**Figure 4 ijms-26-08505-f004:**
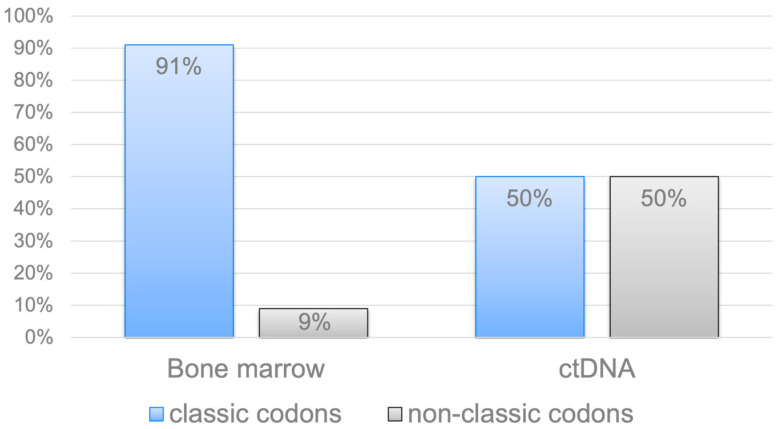
Classical and non-classical codons in the *KRAS* and *NRAS* genes, depending on the tumor location in MM. Rare *KRAS* and *NRAS* gene mutations were significantly more frequently detected in tumor ctDNA.

**Figure 5 ijms-26-08505-f005:**
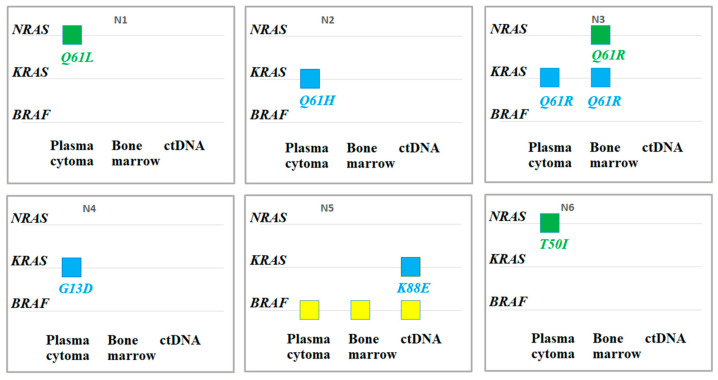
Mutations in the RAS–ERK pathway genes in cases where three loci were investigated. Mutations identified in plasmacytoma rarely matched the bone marrow or ctDNA substrate.

**Figure 6 ijms-26-08505-f006:**
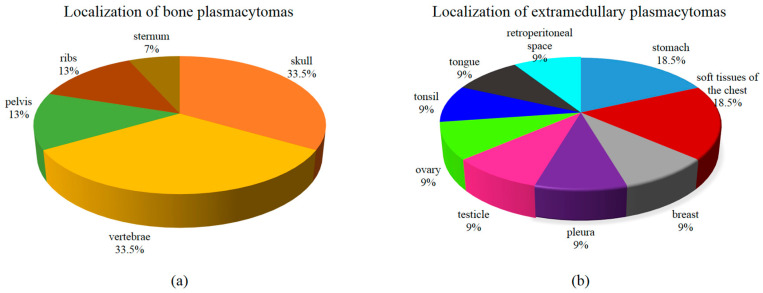
Localization of plasmacytoma: (**a**) Bone; (**b**) Extramedullary.

**Figure 7 ijms-26-08505-f007:**
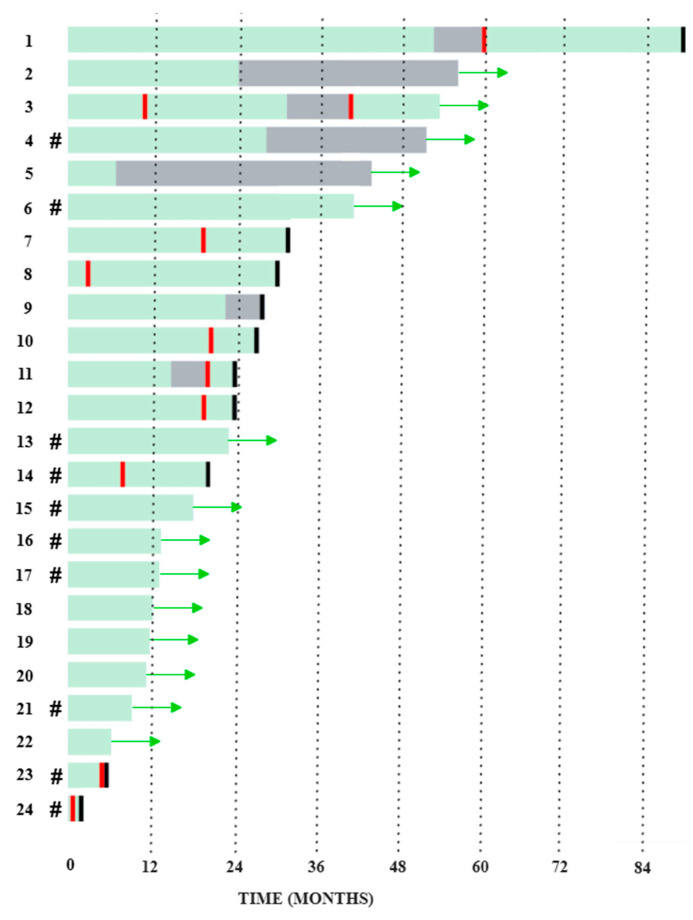
Follow-up of 24 patients with MM complicated by plasmacytoma. Black indicates death, red—relapse, gray—observation without therapy, green—continuation of therapy. A green arrow indicates continuation of observation of the patient, # symbol indicates the presence of mutations in the RAS–ERK pathway genes in plasmacytoma.

**Figure 8 ijms-26-08505-f008:**
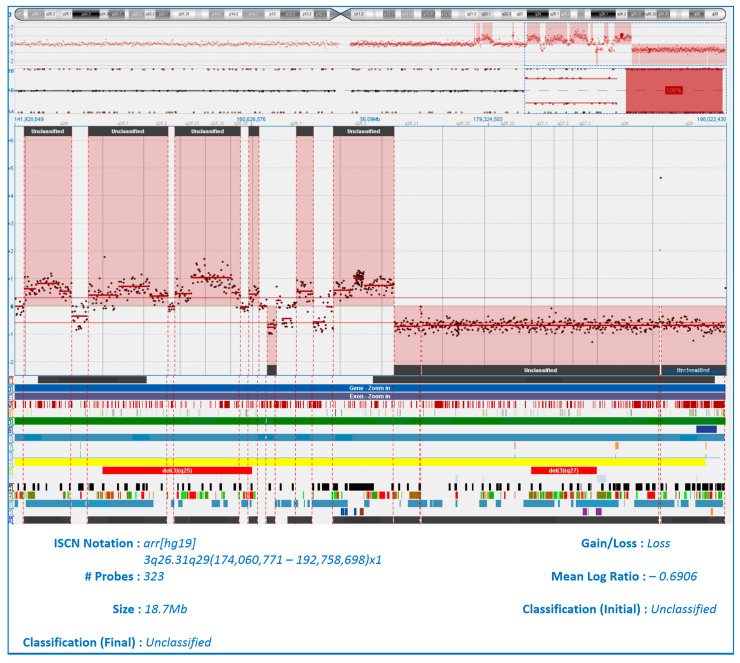
Сhromothripsis of 3q in the substrate of extramedullary plasmacytoma (Patient No. 12).

**Figure 9 ijms-26-08505-f009:**
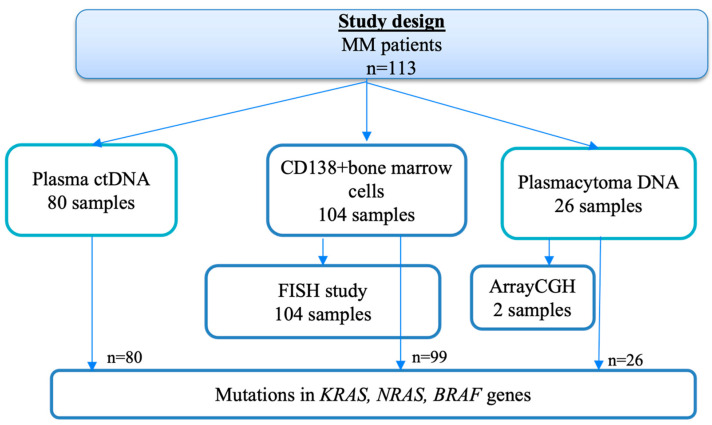
Study design.

**Table 1 ijms-26-08505-t001:** Analysis of certain parameters in two cohorts of patients with MM.

Parameters	Patients with Plasmacytomas	Patients Without Plasmacytomas	*p*
	(*n* = 50)	(*n* = 30)	
Age, years, median, and range	56.5 (41–73)	55 (29–83)	0.85
Male/female	22 (44%)/28 (56%)	14 (47%)/16 (53%)	0.82
Type of secretion			0.88
G	30 (60%)	16 (54%)	
A	10(20%)	9 (30%)	
BJ	5 (10%)	3 (10%)	
D	3 (6%)	1 (3%)	
FLC	2 (4%)	1 (3%)	
BJ protein excretion			0.63
Yes	31 (62%)	21 (70%)	
No	19 (38%)	9 (30%)	
Type of FLC			0.1
κ	32 (64%)	17 (57%)	
λ	18 (36%)	13 (43%)	
D-S stage			<0.0001
IA, IB	1 (2%)	5 (17%)	
IIA	3 (6%)	8 (27%)	
IIIA	37 (74%)	14 (47%)	
IIIB	9 (18%)	3 (10%)	
ISS stage			0.94
I	16 (32%)	10 (33%)	
II	11 (22%)	8 (27%)	
III	9 (18%)	4 (13%)	
ND	14 (28%)	8 (27%)	
Hemoglobin (g/L), median, and range	115 (66–148)	108.5 (72–156)	0.39
LDH (U/L), median, and range	182.5 (65 -694)	166.5 (71–385)	0.25
Plasma cells in bone marrow aspiration, %, median, and range	17.6 (0–92.0)	15.1 (4.0–56.0)	0.95
FISH			0.68
Standard risk	21 (42%)	16 (53%)	
High risk	28 (56%)	14 (46%)	
ND	1 (2%)	0	
Double/Triple hit (*n* = 42)			0.69
Yes	7 (25%)	2 (14%)	
no	21 (75%)	12 (86%)	

**Table 2 ijms-26-08505-t002:** *KRAS*, *NRAS*, and *BRAF* gene mutations in two cohorts of patients with MM.

Mutations in Genes	Patients with Plasmacytomas (*n* = 50)	Patients Without Plasmacytomas (*n* = 30)	*p*
In bone marrow			
*KRAS*	6 (12%)	7 (23%)	0.2
*NRAS*	11 (22%)	2 (7%)	0.12
*BRAF*	7 (14%)	2 (7%)	0.5
*KRAS* + *NRAS*	1 (2%)	0	1
*NRAS* + *BRAF*	1 (2%)	1 (3%)	1
In ctDNA			
*KRAS*	5 (10%)	-	0.15
*NRAS*	5 (10%)	-	0.15
*BRAF*	4 (8%)	-	0.29

**Table 3 ijms-26-08505-t003:** Molecular karyotype lesions common for patients 12 and 24.

Chromosomal Localization	Type of Aberration	Size, Mb ^1^
1p34	del	4.67–25.59
1q21	gain	8.41–22.36
10q11	del	92.79
16q11.2	del	32.55–43.54
17p13	del	8.05–18.89
Xp22	del	58.37
Xq11	del	25.18–92.89

^1^ Lesions of sizes less than 4.5 Mb are not included.

**Table 4 ijms-26-08505-t004:** Characteristics of MM patients included in the study.

Parameters	Patients with MM (*n* = 113)
Age, years, median, and range	55 (29–83)
Male/female	48/65
Type of secretion	
G	69 (61%)
A	23 (20.4%)
BJ	15 (13.3%)
D	4 (3.5%)
non-secretory	2 (1,8%)
Type of FLC	*n* = 111
κ	68 (61%)
λ	43(39%)
D-S stage	*n* = 112
IA, IB	6 (5.3%)
IIA	19 (17%)
IIIA	68 (60.7%)
IIIB	19 (17%)
ISS stage	*n* = 79
I–II–III	46%–28%–26%
Hemoglobin (g/L), median, and range	110 (55–156)
LDH (U/L), median, and range	174 (65–694)
Plasma cells in bone marrow aspiration, %, median, and range	16 (0.4–58)
FISH	*n* = 104
Standard risk	48 (46%)
High risk	56 (54%)
Plasmacytomas	
yes	74 (65%)
no	39 (35%)
	*n* = 74
bone (B)	59 (80%)
extramedullary (E)	5 (7%)
B + E	10 (13%)

## Data Availability

Data are contained within the article and Appendix A.

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
