# Peer review of "Liquid Biopsy as a Means of Assessing Prognosis and Identifying Novel Risk Factors in Multiple Myeloma"

_ijms, 2025, doi:10.3390/ijms26178505_

Round 1

Reviewer 1 Report

Comments and Suggestions for Authors

The manuscript presents a complete, well-organized, and scientifically relevant study of liquid biopsy and ctDNA analysis for the identification of RAS-ERK pathway mutations in multiple myeloma (MM), particularly related to plasmacytomas. The study provides novel data regarding spatial genetic heterogeneity, and the usefulness of ctDNA as a non-invasive diagnostic and prognostic tool. However, there are a few points that should be clarified and less need to be modified to advance the purpose of this manuscript.

Major Comments:

  1. Please provide the statistical tests used for each comparison, and how multiple comparisons were controlled for. Some subgroup sample sizes (for example the plasmacytoma samples n=11) are small and will decrease power.
  2. Although survival data is presented in figure 7, there are no survival curves or statistically testing (Kaplan-Meier, Cox regression) to estimate whether mutations found in ctDNA have any significant prognostic impact compared to mutations from other compartments.
  3. The chronotherapies results (figure 8) are also interesting, but there is no investigation of mechanisms.
  4. The discussion is missing mention of whether the identified markers could drive therapeutic decisions, or how they could, especially with new RAS-pathway inhibitors, which should be noted.

Minor Comments

  1. There were some references that were not referenced correctly or are referenced as “[Error! Reference not found.]” — this must be remedied.
  2. Some acronyms are identified that are not defined on first use (cnLOH, for example, in Table 4).
  3. Figures 3-5 are good, but larger font and more distinct legends would benefit the figures, particularly those not familiar with MM cytogenetics.

Author Response

Comments 1.  Please provide the statistical tests used for each comparison, and how multiple comparisons were controlled for. Some subgroup sample sizes (for example the plasmacytoma samples n=11) are small and will decrease power.

Response 1. To test hypotheses about differences in the distributions of categorical features in the comparison groups, the two-sided Fisher criterion was used. To test hypotheses about the presence of differences in the distributions of numerical indicators in the comparison groups, the nonparametric Mann-Whitney rank criterion was used. We did not conduct multiple comparisons; specific single hypotheses were tested, so we did not adjust for the multiplicity of hypotheses. Yes, we agree with you that the sample of plasmacytomas is small, so we draw conclusions about the mutational status of genes in plasmacytomas very carefully.

Comments 2. Although survival data is presented in figure 7, there are no survival curves or statistically testing (Kaplan-Meier, Cox regression) to estimate whether mutations found in ctDNA have any significant prognostic impact compared to mutations from other compartments.

Response 2. We used event-based analysis methods with Kaplan-Meier estimates and log-rank test to assess statistical significance of differences. In the group of 24 patients shown in Figure 7, we attempted to analyze overall survival and progression-free survival depending on the presence of mutation in plasmacytoma, but did not obtain an interpretable result due to insufficient sample power. We did not plot survival for the entire group because the group is heterogeneous and includes both candidates and non-candidates for transplantation. In the next article, we plan to analyze the treatment efficacy indicators in a more homogeneous group of patients. Mutations were more often detected in bone marrow, and cases of detection of mutation in ctDNA without confirmation in bone marrow were extremely rare. Therefore, in our study, we did not analyze the additional prognostic significance of detecting mutations only in ctDNA.

Comments 3. The chronotherapies results (figure 8) are also interesting, but there is no investigation of mechanisms.

Response 3. We were also very interested to see chromothripsis in a patient with MM, whose disease was extremely aggressive. We do not perform molecular karyotyping in routine practice, so we have not encountered such a phenomenon in myeloma. We believe that such a finding requires a separate approach to analysis. Perhaps, in some cases (double / triple hit myeloma, myeloma with extramedullary plasmacytoma, plasma cell leukemia) it is necessary to perform a molecular karyotype of the tumor to search for chromothripsis and chromoplexia and concentrate on data analysis.

Comments 4. The discussion is missing mention of whether the identified markers could drive therapeutic decisions, or how they could, especially with new RAS-pathway inhibitors, which should be noted.

Response 4. Thank you, this is really important. Information added (Lines 262-265)

Comments 5. There were some references that were not referenced correctly or are referenced as “[Error! Reference not found.]” — this must be remedied.

Response 5. Thank you, fixed

Comments 6. Some acronyms are identified that are not defined on first use (cnLOH, for example, in Table 4).

Response 6. Thank you, fixed (Line 228)

Comments 7. Figures 3-5 are good, but larger font and more distinct legends would benefit the figures, particularly those not familiar with MM cytogenetics.

Response 7. Thank you, the figures have been enlarged and additional information has been added to the figure captions.

 Summary. Dear Reviewer, we thank you very much for your analysis and high evaluation of the work, valuable comments and instructions. We have tried to correct everything. In addition, we have tried to improve the English language in the manuscript. We understand the need to increase the sample of plasmacytomas to make the conclusions reliable. With great respect, the authors

Reviewer 2 Report

Comments and Suggestions for Authors

I’d like to thank the authors for their submission to IJMS. This article reflects an important study relevant to multiple myeloma prognostication. Detection of KRAS, NRAS, and BRAF mutations in ctDNA may offer greater sensitivity in some cases verses bone marrow biopsy, especially in cases involving extramedullary plasmacytomas and/or in the case of non-classical mutations. Liquid biopsy is also less invasive and thus preferable in an already vulnerable patient population. This could represent a valuable tool in the clinic for prognostication, guidance of treatment decisions, and detection of minimal residual disease following therapy. There are some typographical and formatting errors which need to be addressed, and there are some gaps in the analysis that make drawing conclusions challenging. We need to know in these paired samples, the rate of concordance between BM samples and ctDNA from plasma. Otherwise, we cannot reliably evaluate the differences in these sampling methods to draw certain conclusions. Please refer to my comments below for recommendations:

Lines 36-37: Fix incomplete sentence and reference/source.

Line 60: and a decrease in sensitivity…

Lines 78-79: in the RAS-ERK genes in different MM substrates…

Figure 1: To improve quality, subtract out the grey background from the plot.

Figure 3: What is the percentage of overlap of samples with mutations detected in bone marrow and ctDNA? It seems like ctDNA is less sensitive than BM biopsy in this Figure. Is that true? If so, and there is a lack of concordance between BM and ctDNA, do we think we are measuring something distinct in ctDNA verses in BM? If there is concordance between BM and ctDNA mutation detection, why would we utilize ctDNA over BM which appears for sensitive?

Figure 4: Same comment. What is the measure of overlap/concordance between BM and ctDNA samples? Perhaps the ctDNA measures a subpopulation of non-classical mutants undetected in BM samples?

Figure 5: It seems that measurements of mutations depend on sample location and that there is a lack of concordance between sample locations. It could be that each sample (ctDNA vs BM vs plasmacytoma) is measuring a different compartment of disease, and that it may be necessary to evaluate all 3 compartments to get a full picture of disease mutation burden. The data in N3 seem to suggest that ctDNA is not sensitive to the presence of mutations detected in plasmacytoma or BM. Whereas, N5 shows a variant detected in ctDNA exclusively. Should we conclude that for plasmacytoma patients, we must perform liquid biopsy in addition to BM and plasmacytoma biopsies? Again, do BM and ctDNA from plasma measure the same thing, or are they distinct? In N6 it appears that ctDNA is missing. Is this real or a formatting issue?

Figure 6: Please flip the charts a and b so that 6(a) is left of 6(b). Please also do not recycle colors between charts. I.e. pick a different, unique color for each location. Otherwise, it may lead to confusion in the interpretation of results. Skull and stomach should not both be blue.

Line 174-175: Of the 15 bone plasmacytoma mutations in the RAS-ERK genes, KRAS mutations were detected in 27% of cases…

Table 3 & lines 180-182: What statistical test was applied to determine the difference between bone and extramedullary plasmacytoma mutation frequency? Where are your p-values?

Line 184: Capitalize Figure 7.

Figure 7: In line 177 and 180 you state that 11 cases had mutations in extramedullary plasmacytoma, but there are only 10 patients marked with # in Figure 7. Is this an omission due to error, or are we looking at two different populations? Also, please remove the grey background from the figure to enhance clarity and aesthetics. Consider adding a vertical line(s) or grid for ease of interpretation.

Line 223-224: You state there are similar lesions in chromosome 7, but do not appear to list these in Table 4.

Line 369: What version of the CytoSure was used and who is it produced/published by?

Overall: This study utilizes a reasonably large cohort of MM patients with plasmacytoma. There are a few areas where the descriptions and figures could be improved to better support the conclusions. It is important to note/clarify the level of overlap and concordance between samples to support drawing accurate conclusions about mutation frequencies/propensities. The data seem to suggest a multifaceted approach involving multiple biopsies (including liquid biopsy) and multiple assays are required to fully appreciate the mutational burden of a given MM patient complicated by plasmacytoma. I would agree that there is sufficient data presented to recommend that BM biopsy alone may not be sufficient. As liquid biopsy analysis methods and protocols improve in sensitivity, we may find more sensitive assays/platforms give us more complete information. In the meantime, liquid biopsy may be an important adjuvant to BM and plasmacytoma biopsy when it comes to evaluating disease progression, response, and prognosis.

Author Response

Comments 1. Lines 36-37: Fix incomplete sentence and reference/source.

Response 1.  Thank you, fixed

Comments 2. Line 60: and a decrease in sensitivity… 

Response 2. Thank you, fixed

Comments 3. Lines 78-79: in the RAS-ERK genes in different MM substrates… 

Response 3. Thank you, fixed

Comments 4.  Figure 1: To improve quality, subtract out the grey background from the plot. 

Response 4. Thank you, fixed

Comments 5 and 6. Figure 3: What is the percentage of overlap of samples with mutations detected in bone marrow and ctDNA? It seems like ctDNA is less sensitive than BM biopsy in this Figure. Is that true? If so, and there is a lack of concordance between BM and ctDNA, do we think we are measuring something distinct in ctDNA verses in BM? If there is concordance between BM and ctDNA mutation detection, why would we utilize ctDNA over BM which appears for sensitive?

Figure 4: Same comment. What is the measure of overlap/concordance between BM and ctDNA samples? Perhaps the ctDNA measures a subpopulation of non-classical mutants undetected in BM samples?

Response 5 and 6. Thank you for these valuable comments. Below is the analysis (Attached is a document with a concordance figure). The KRAS mutation was detected both in bone marrow and ctDNA for 2 patients (2%), only in bone marrow without confirmation in ctDNA for 12 patients (15%), only in ctDNA without confirmation in bone marrow for 3 patients (4%). The NRAS mutation was detected both in bone marrow and ctDNA for 4 patients (5%), only in bone marrow without confirmation in ctDNA for 12 patients (15%), only in ctDNA without confirmation in bone marrow for  1 patient (1%). The BRAF mutation was detected both in bone marrow and ctDNA for 3 patients (4 only in bone marrow without confirmation in ctDNA for 8 patients (10%), %), only in ctDNA without confirmation in bone marrow for 1 patient (1%).  Mutations had been detected in bone marrow more frequently, and cases of detecting a mutation in ctDNA without confirming in bone marrow were extremely rare. 

You are right, based on the presented concordance coefficient, ctDNA is less sensitive than bone marrow. In connection with this, in our opinion, it is not possible to use liquid biopsy as a method for assessing minimal residual disease in multiple myeloma. We believe that in different anatomical locations we find different populations/subclones of tumor cells. It is quite possible that we see more aggressive cells in circulating plasma and plasmacytomas. Bone marrow with its microenvironment is an place familiar to plasma cells. Presumably, for a plasma cell to lose its homing mechanism and leave the bone marrow, appearing in the blood and forming plasmacytomas, it must have a more aggressive phenotype. This probably explains the fact that non-classical mutations are detected significantly more often in ctDNA than in bone marrow. We have noticed that individuals with rare mutations have many different features of high-risk myeloma.

The fact that in the presence of plasmacytomas we detect mutations in ctDNA, and in the absence of plasmacytomas there are no mutations in ctDNA, dictates the need to focus specifically on the group of patients with plasmacytomas and study this group of patients using liquid biopsy, without wasting resources and time on the second group.

Comments 7. Figure 5: It seems that measurements of mutations depend on sample location and that there is a lack of concordance between sample locations. It could be that each sample (ctDNA vs BM vs plasmacytoma) is measuring a different compartment of disease, and that it may be necessary to evaluate all 3 compartments to get a full picture of disease mutation burden. The data in N3 seem to suggest that ctDNA is not sensitive to the presence of mutations detected in plasmacytoma or BM. Whereas, N5 shows a variant detected in ctDNA exclusively. Should we conclude that for plasmacytoma patients, we must perform liquid biopsy in addition to BM and plasmacytoma biopsies? Again, do BM and ctDNA from plasma measure the same thing, or are they distinct? In N6 it appears that ctDNA is missing. Is this real or a formatting issue?

Response 7. We completely agree with you, it seems that each sample is unique and carries important information. Although we see that liquid biopsy helps to study MM complicated by plasmacytoma, nevertheless, the biopsies of plasmacytomas themselves remain a very valuable impact. Yes, we believe that we need liquid biopsy to understand the pathogenesis of plasmacytoma. With patient N6 the problem of formatting issue was clearly visible, now it is eliminated, thank you

Comments 8. Figure 6: Please flip the charts a and b so that 6(a) is left of 6(b). Please also do not recycle colors between charts. I.e. pick a different, unique color for each location. Otherwise, it may lead to confusion in the interpretation of results. Skull and stomach should not both be blue.

Response 8. Thank you, fixed

Comments 9. Line 174-175: Of the 15 bone plasmacytoma mutations in the RAS-ERK genes, KRAS mutations were detected in 27% of cases…

Response 9. We have slightly rephrased the sentences for better comprehension of the text. Of the 15 bone plasmacytoma samples, mutations were found in 6, of the 11 extramedullary plasmacytoma samples, mutations were found in 5.(Lines 183-187).

Comments 10. Table 3 & lines 180-182: What statistical test was applied to determine the difference between bone and extramedullary plasmacytoma mutation frequency? Where are your p-values?

 Response 10. Thank you. This is probably our mistake. The table simply listed the localizations of plasmacytomas and mutations in a particular gene. Since this information is not of great importance, we removed the table and slightly changed the text. Statistical analysis was not performed, the sample is small, but so far the trend is that in bone plasmacytomas we see a mutation in the KRAS gene more often, and in extramedullary ones - in the NRAS gene. In the future, we plan to analyze a larger sample of patients.

Comments 11. Line 184: Capitalize Figure 7.

 Response 11. Thank you, fixed(Line 192)

Comments 12. Figure 7: In line 177 and 180 you state that 11 cases had mutations in extramedullary plasmacytoma, but there are only 10 patients marked with # in Figure 7. Is this an omission due to error, or are we looking at two different populations? Also, please remove the grey background from the figure to enhance clarity and aesthetics. Consider adding a vertical line(s) or grid for ease of interpretation.

Response 12.Thank you. No, a little differently: out of 11 extramedullary plasmacytoma mutations were in 5 samples. Out of 15 bone plasmacytoma, mutations were found in 6 samples. In total, 11 plasmacytoma samples had mutations. However, one patient had two plasmacytoma biopsies, and a mutation was found in the both biopsies. Accordingly, there were 10 patients with a mutated gene status, they were marked with #. Thank you for correcting the figure, we tried to follow the recommendations, removed the background and added lines.

Comments 13. Line 223-224: You state there are similar lesions in chromosome 7, but do not appear to list these in Table 4.

Response 13. We meant that similar lesions were found at seven distinct chromosomal loci, we corrected the phrase (lines 230-232)

Comments 14.  Line 369: What version of the CytoSure was used and who is it produced/published by?

 Response 14.Thank you. Information added.

Summary. Dear Reviewer,

we sincerely thank you for your deep analysis of our work and its high assessment. Your comments are very accurate and your recommendations are useful. It really seems that each tumor sample has its own disease compartment, which means that the value of analyzing different tumor samples increases. We tried to implement all the recommendations, and also tried to improve the English language of the manuscript

With great respect, the authors
